# Influenza Vaccination Campaign during the COVID-19 Pandemic: The Experience of a Research and Teaching Hospital in Milan

**DOI:** 10.3390/ijerph18115874

**Published:** 2021-05-30

**Authors:** Pier Mario Perrone, Giacomo Biganzoli, Maurizio Lecce, Emanuela Maria Campagnoli, Ambra Castrofino, Andrea Cinnirella, Federica Fornaro, Claudia Gallana, Francesca Maria Grosso, Manuel Maffeo, Viktoriia Shishmintseva, Elena Pariani, Silvana Castaldi

**Affiliations:** 1Department Biomedical Sciences for Health, Postgraduate School in Public Health, University of Milano, 20136 Milan, Italy; emanuela.campagnoli@unimi.it (E.M.C.); ambra.castrofino@unimi.it (A.C.); andrea.cinnirella@unimi.it (A.C.); federica.fornaro@unimi.it (F.F.); claudia.gallana@unimi.it (C.G.); francesca.grosso@unimi.it (F.M.G.); manuel.maffeo@unimi.it (M.M.); viktoriia.shishmintseva@unimi.it (V.S.); Elena.Pariani@unimi.it (E.P.); Silvana.Castaldi@unimi.it (S.C.); 2Pharmacogenomics & Precision Therapeutics Master Degree, University of Milan, 20142 Milan, Italy; giacomo.biganzoli@studenti.unimi.it; 3Quality Unit Fondazione IRCCS Ca’ Granda OMP, 20122 Milan, Italy; Maurizio.lecce@studenti.unimi.it

**Keywords:** influenza vaccine, vaccination adherence, healthcare workers vaccination, vaccination refusal

## Abstract

Background: During the COVID-19 pandemic, more than ever, optimal influenza vaccination coverage among healthcare workers (HCWs) is crucial to avoid absenteeism and disruption of health services, as well as in-hospital influenza outbreaks. The aim of this study is to analyze the 2020 influenza vaccination campaign, comparing it with the previous year’s in a research and teaching hospital in Northern Italy. Methods: adopting an approach based on combined strategies, three interventions were deployed: a promotional and educational campaign, vaccination delivery through both ad hoc and on-site ambulatories, and a gaming strategy. Personal data and professional categories were collected and analyzed using univariate logistic regression. Vaccinated HCWs were asked to fill in a questionnaire to describe their reasons for vaccination adherence. Results: the vaccination coverage rate (VCR) was 43.1%, compared to 21.5% in 2019. The highest increase was registered among administrative staff (308.3%), while physicians represent the most vaccinated category (n = 600). Moreover, residents (prevalence ratio (PR): 1.12; 95% CI 1.04–1.20), as well as intensive care (PR: 1.44; 95% CI: 1.24–1.69) and newborn workers (PR: 1.41; 95% CI: 1.20–1.65) were, respectively, the categories most frequently vaccinated for the first time. Conclusion: the significant increase in vaccination coverage rate confirms the suitability of the combined strategy of delivering the flu vaccination campaign and represents a first step towards reaching WHO recommended vaccination rates.

## 1. Introduction

Seasonal influenza is an infectious disease with a high impact on public health systems, both in terms of incidence and in terms of morbidity and mortality [1,2,3]. The World Health Organization (WHO) estimates that, worldwide, the annual influenza epidemic results in 3–5 million cases of severe illness and 290,000 to 650,000 deaths [4]. In Europe, seasonal influenza accounts for 4–50 million cases and 15,000–70,000 deaths every year [5]. Healthcare workers (HCWs) are exposed to an increased risk of contracting flu and spreading it to vulnerable patients, colleagues, and relatives when compared to the general population [6,7]. At the same time, influenza infection of HCW could lead to an increase in absenteeism, causing economic losses [8]. Therefore, they represent a major target group for vaccination, with the aim of preventing severe disease in themselves, absenteeism, and the consequent disruption of healthcare services, as well as to avoid hospital outbreaks among patients–critical considerations at a time when healthcare systems around the world are struggling against COVID-19 [9,10]. Despite the fact that most international public health associations promote influenza vaccination as a major preventive strategy [11,12], the vaccination coverage rate (VCR) is still very low among HCWs. According to the latest estimates, in Europe the median VCR among HCWs was 30.2% during the 2016–2017 influenza season, with the highest rates being reported in Belgium, England, and Wales, while the lowest rates were reported in Norway and Italy [13]. Focusing on Italy, despite a recommended VCR target of 75% [11], the observed rate amounts to only 15.6% [13]. The aim of this study is to evaluate the compliance of HCWs with influenza vaccination in IRCCS Fondazione Ca’ Granda Ospedale Maggiore Policlinico (hereafter Fondazione-OMP), a research and teaching hospital in Milan, Italy, to compare the VCR of this year to that of previous years and to analyze adherence to vaccination in different population subgroups.

## 2. Materials and Methods

The 2020 influenza vaccination campaign for HCWs in Fondazione-OMP was carried out from 26 November to 22 December 2020. According to the most recent studies in literature, which highlight that an optimal flu vaccination coverage among HCWs is more likely to be achieved by combining multiple strategies [14,15], the working group decided to deploy three interventions: a promotional and educational campaign, vaccination delivery through both an ad hoc and on-site ambulatory, and the triggering of a competition among hospital departments (so called gaming strategy).

The promotional and educational campaign, introduced for the first time this year, started in September 2020 and was conducted using the hospital intranet platform. Every HCW could access a dedicated online page where the start of the flu vaccination campaign was widely circulated, and operational information for accessing vaccinations was prominent. This online environment was designed with an attractive graphic layout. The promotional section was based on a survey conducted in the 2019–2020 vaccination campaign investigating reasons for vaccination refusal among HCWs, and consisted of several claims which emphasized the reasons for and benefits of vaccination for this target group. The educational section was specifically dedicated to a myth debunking intervention, contrasting the most common misconceptions about flu and flu vaccination (e.g., “I am young and I don’t think I will catch influenza”).

As in previous campaigns, the core strategy consisted of offering vaccination through both an ad hoc and on-site ambulatory [16]. The ad hoc ambulatory was placed in the same location throughout the campaign and was open every day from 10 a.m. to 4 p.m. (15 days in total). A team of three vaccinators was constantly present, two of whom were involved in vaccine administration—with the third one fulfilling a supporting role. Access to the ad hoc ambulatory was by appointment. Each HCW could book his slot via the hospital intranet platform, with up to a maximum of 180 bookable slots being available per day. Otherwise, the on-site ambulatory consisted of teams of hygiene and preventive medicine residents visiting each hospital building at least once and if possible, twice, from 8 a.m. to 2 p.m. in order to cover two work shifts, thus intercepting as many HCWs as possible directly in their own working place. Vaccination teams were made up of two or three residents, based on the size and characteristics of each building to avoid personnel queues.

The gaming strategy, also introduced for the first time, consisted of triggering a competition among the eight departments into which Fondazione-OMP is functionally and logistically divided. The underlying rationale was that every HCW would get his flu shot to make his department win, starting a virtuous mechanism that would lead to a boost in the VCR of the entire hospital. The intranet platform played a major role again, including a section where a daily updated ranking of the departments was displayed.

Vaccinated HCWs were asked to fill in a form that collected data, including personal data, professional category (physician, nurse, student, administrative, researcher, or other), specialty, influenza vaccination history, brief medical history to assess risk category, and informed consent.

An anonymous and self-administered questionnaire was used to assess the reasons for adherence to vaccination as well as age, gender, professional category and area of activity. Fondazione-OMP includes all medical specialties, while for surgery specialties, it has general surgery, urology, otorhinolaryngology, ophthalmology, neurosurgery, vascular, thoracic, maxillofacial departments, and a transplant unit for liver, lung, and kidney; every specialty was reached in order to maximize personnel coverage.

According to the recommendations of regional health authorities, the administered vaccine was Flucelvax Tetra^®^, an inactivated tetravalent antigen vaccine produced by Seqirus Netherlands B.V. (Amsterdam, The Netherlands), injected intramuscularly into the deltoid muscle.

All collected data were registered in a regional database managed by ATS Milan.

All the statistical analyses were conducted with R software, release 4.0.0. A newly generated variable (age) was computed by considering the date of birth and the date of the vaccination for every observation. A summary statistic reporting median and interquartile range was produced and subsequently categorized into classes based on these specific intervals: 18–39; 40–59; 60–80. For each class, the number of subjects and the percentages were reported. Other variables were categorical, and for each class the number of subjects and percentage were reported. Firstly, mosaic plots were produced to graphically explore the relationships between the vaccination site and the demographic characteristics, occupation, and area of activity of the subjects [17,18]. Then, the data were analyzed using univariate logistic regression with a log link, and model results were reported as prevalence ratios with a 95% confidence interval. The likelihood ratio was used to test the general association between the vaccination site and the other categorical variables. The same analysis was performed to explore and assess the relationships between the attitude towards previous vaccination, demographic characteristics, occupation, and area of activity of the subjects.

No ethical approval was required for this study, according to Regional Law No.3 of 2012 of the Lombardy Region.

## 3. Results

Table 1 shows the population characteristics of the total vaccinated subjects. Out of 2103 vaccinated HCWs, 1418 (67.5%) were female. Regarding age, the median was 43, with first and third quartiles equal to 31 and 54, and an interquartile range of 23. The most represented age group is the 18–39 age class, made up of 938 operators (44.6%), immediately followed by the 40–59 age group, made up of 925 operators (44%). The least represented age group is the 60–80 years old class, with 240 vaccinated operators (11.4%). Regarding the professional category, physicians constitute the largest group, equal to 600 subjects (28.5%). Nursing staff follows with 452 vaccinated operators (21.5%). Concerning the areas of activity, the highest number of vaccinated subjects was found in specialist medicine (34%), followed by specialist surgery (13.6%) and administration (13.1%). It is practicable to compare the population of HCWs vaccinated this season to the population of HCWs vaccinated in the previous campaign (2019–2020). The gender distribution substantially overlaps, since of the 1153 operators vaccinated in the 2019–2020 season, females are only slightly less represented compared to the current season (64.2% vs. 67.5%) while the opposite occurs for males (35.8% vs. 32.5%). The population vaccinated in the previous season was on average younger (median 36, interquartile range 25) than the current season (median 43, interquartile range 23). Of note is the clear preponderance of the 18–39 age group in the 2019–2020 vaccinated population, corresponding to 55.7% of all those vaccinated. The 40–59 age group last year only represented 32.4% of those vaccinated. This difference has leveled off in the current season, in which the two age groups have reached fairly similar prevalence percentages (44.6% for 18–39 age group and 44.0% for 40–59 age group). The 60–80 age group shows a comparable percentage between the two seasons (11.9% vs. 11.4%). With regard to professional category, physicians were more represented in the current vaccination campaign than in the previous one (28.8% vs. 24.9%). The number of physicians vaccinated has more than doubled (600 vs. 283), however the highest variation is found among administrative staff (+308.3%), followed by auxiliary staff (+252.9%). Regarding the area of activity, the highest increases, more than 100%, were observed in the administrative area (close to a tenfold increase, from 27 to 275 HCWs vaccinated (+918.5%)), the intensive care unit area (more than doubled, from 72 to 183 HCWs vaccinated (+154.2%)), and the pediatric area (also doubled, from 117 to 245 HCWs vaccinated (+109.4%)).

The vaccination forms enabled us to conduct a detailed analysis of HCWs, focusing on two issues. In order to understand the impact of the implemented strategies, we looked at the difference between on-site and ad hoc ambulatory groups, and the difference between the ‘previously vaccinated’ and ‘never vaccinated before’ groups.

Figure 1 shows the distribution of area of activity in the ad hoc and on-site ambulatory, respectively. The bar chart with expressed percentages enables an immediate comprehension of this difference. Analyzing the ad hoc ambulatory rates, we can see that there are high percentages in three areas, namely the administrative, the general medicine, and the intensive care unit areas. On the contrary, the newborn, specialist surgery, and specialist medicine areas had the highest rates at the on-site ambulatory.

In Figure 2, the same results can be seen through a mosaic plot, with a more comprehensible graphic layout: specifically, the administrative and specialist medicine areas show opposite ratios regarding the choice of the type of ambulatory.

In Figure 1 and Figure 2, the area of activity is reported because this represents the main feature showing significant PR.

Table 2 shows the results of the logistic regression model: the prevalence ratio compares on-site versus ad hoc ambulatory with a 95% confidence interval and likelihood ratio test.

Comparing to physicians (reference category), residents (PR: 1.06; CI: 0.99–1.13) and administrative staff (PR: 1.03 CI: 0.96–1.11) represent the professions most likely to be vaccinated at the on-site ambulatory.

The area of activity analysis, considering the administrative area as reference area, shows the highest prevalence ratio in the specialist medicine (PR: 1.26; 95% CI: 1.16–1.37) and newborn areas (PR: 1.24; 95% CI: 1.12–1.37).

As can be seen, there are no significant differences between sex (*p* = 0.76641) and age (*p* = 0.08831) that might affect on-site or ad hoc ambulatory choice.

Graphic representations of the correlation between specific area of activity and ‘previous vaccination’ vs. ‘first time vaccination’ are shown in Figure 3 and Figure 4. In Figure 3, among the never vaccinated before, the most represented area is specialist medicine (36.26%), followed by administration (16.76%). Interestingly, the specialist medicine area is also the most represented area in the previously vaccinated population.

The same results, from an even more intuitive graphic point of view, are expressed in Figure 4. As seen for previous graphic representations, Figure 3 and Figure 4 report the area of activity, because this represents the main feature showing significant PR.

Table 3 compares the populations of healthcare workers who first received the flu vaccine this season and those who have already received the vaccination in the past, showing the results of the logistic regression model with a 95% confidence interval and likelihood ratio test. The analysis shows the highest prevalence ratio for the intensive care unit (PR: 1.44; 95% CI: 1.24–1.69) and the newborn area (PR: 1.41; 95% CI: 1.20–1.65) compared to the administrative area, which was chosen as a reference. Moreover, taking physician as a reference profession, residents show the highest prevalence ratio (PR 1.12; 95% CI: 1.04–1.20).

A higher prevalence of male sex in the never vaccinated before group should also be noted (PR: 1.12; 95% CI: 1.05–1.21).

As can be seen, there are no significant differences between age groups (*p* = 0.08831) regarding previous vaccination status.

Table 4 shows the results of the survey about reasons for vaccination adherence. The total number of questionnaires collected was 1899. A significant difference was observed between the genders, with a greater proportion of female HCWs (64.2%) responding to the questionnaire. The two age intervals most represented are 18–39 and 40–59, with only a slight difference between them (41.3% and 41.2%, respectively). This could be explained by the high number of residents immunized during the vaccination campaign, a professional category within which many are under 35 years of age. Since it was possible to state more than one reason for vaccination, the total number of answers is greater than the total number of questionnaires received. The most frequent reason for vaccination is the belief that vaccination is an effective disease prevention tool, which was checked by 1437 HCWs (75.7%). Being HCWs is an important factor in choosing whether to receive a vaccination, since 806 HCWs (42.4%) got vaccinated to protect their patients against the virus, while 359 HCWs (18.9%) considered themselves to be more exposed to influenza than the general population.

## 4. Discussion

Due to a structured promotional campaign and a well-structured strategy for vaccine delivery all around the hospital, our working group reached a vaccination coverage close to 43.1%, with a particularly noticeable increase in female HCWs (+91.6%), administrative workers (+918.5%), and in HCW aged between 40 and 60 years (+147.3%) getting vaccinated.

Although this increase in vaccination coverage is close to double the previous campaign’s results [16], it is still far below the European Council’s recommendation of 75% [19]. This discrepancy could be partially explained by the non-delivery of vaccine supplies, given a dramatic vaccine shortage which affected the entire Lombardy region. Due to this, the ad hoc ambulatory ended up being open 13 days instead of the initially planned 15 days. The schedule of the onsite ambulatory was also affected, leading to the cancellation of many onsite visits, with the result being that at least one visit to each building, and a second visit for those with greater numbers of HCWs. 

The increase in vaccination coverage could be explained by several different elements, such as a targeted promotional and educational campaign based on the analysis of a previous survey on reasons for vaccine hesitancy, the success of the on-site vaccination strategy, or the proposal of a challenge between hospital departments (as suggested by the WHO regional office for Europe) [20].

The analysis of our survey didn’t allow us to point to a clear correlation between the fear of COVID-19 and the increase in vaccination adherence. Nevertheless, Di Pumpo et al. suggests this correlation, noting that the COVID-19 pandemic was the major differentiating factor between the previous and present campaigns [21]. This position may be supported by the increase in vaccine coverage observed in some professional classes not directly involved in taking care of frail patients, such as administrative or auxiliary staff, who showed increases of close to 100% over previous vaccination coverage, or by analyzing the increase in VCR in specific areas of activity not involved directly in medical care, such as the administrative area. This could also be linked with the massive communication campaign, led by mass media, about the critical importance of receiving a flu vaccination this year in order to avoid overwhelming the healthcare system. Despite the considerations surrounding the COVID-19 pandemic and the fear of getting infected by this novel virus, the strategy of vaccine delivery all around the hospital, including directly inside clinical wards, in order to increase the adherence of groups who could have problems reaching a vaccination point, proved to be effective. Categories such as physicians and residents show on-site vaccination rates respectively four and six time higher than ad hoc.

We have also seen a significant increase in on-site vaccination rates for every professional category; this could be explained by several factors: firstly, an increased confidence in this type of vaccination strategy, after its first introduction during 2019–2020 campaign; secondly, an extensive coverage of multiple wards, especially during the first days of the campaign, through the deployment of multiple vaccination teams. It’s important to note that due to COVID-19 related logistical problems, we have not been able to enhance on-site vaccination in order to cover all three working shift. This element should be taken into account, particularly for those professions directly involved in patient care who, working night shifts, might have problems reaching an ambulatory in the late morning.

Regarding age and sex, as previously described, we can see a higher increase of vaccination adherence in female HCWs and an increase of attention toward influenza vaccination especially in HCW aged between 40 and 60 years, in contrast to a multicentric Italian study. It’s interesting to note, also, the differences related to the area of activity, with an adherence in specialist medicine more than ten times higher than that of general surgery, once again in contrast to a multicentric Italian study [22].

Our results about reasons for getting vaccinated show that the leading ones involve confidence with immunization and a responsibility towards patients. These answers may be due to the clinical subgroup of a hospital population directly involved in the cure of frail patients that could be severely affected by influenza and its complications. This survey will be of valuable assistance, since the previous one did not investigate the reasons for acceptance but focused on reasons for refusal. Not only is it better to know about HCWs knowledge and behavior regarding immunization and influenza, but it can also help us to increase the efficacy of next year’s vaccination campaign. During the data analysis, we should keep in mind that approximately 223 workers refused to fill in the questionnaire.

Despite great results in terms of vaccination coverage increase, this study shows several limitations. For instance, it involved a relatively small sample size and was based solely on a single hospital. It was also conducted in Italy, a country with high heterogeneity in vaccine coverage rate [23]. Other limitations could be the inability, due to the COVID-19 situation, of on-site vaccination teams to drive an active recruitment of HCWs similarly to previous campaigns and, linked to this, the inability to carry out a new survey about vaccine hesitancy. This could be a reason for the VCR observed, as in the previous experience of vaccine teams in which an active recruitment (like a door-to-door salesperson) represented a fundamental element in persuading more reticent HCWs. Finally, in order to avoid bias, questionnaires on vaccination adherence motivation were anonymous, as previously described, making it difficult to conduct a cross-analysis between data collected from the vaccination form and data collected from the questionnaire in order to find links between specific area of activity and reasons for vaccination.

## 5. Conclusions

Despite coverage rates falling short of WHO targets, the campaign shows a critical positive change from previous ones. The increase of vaccination coverage could be linked firstly to the analysis of 2019–2020 vaccination and the creation of a promotional and educational campaign based on the strategies assumed in the previous study [16]. Linked to these strategies, we structured tailored advertising designed to break down wrong beliefs about influenza and immunization and we also considered the logistical and organizational problems of a large research and teaching hospital, as suggested by different authors [24,25]

Clearly the COVID-19 pandemic has changed several elements in relation to vaccines and vaccinations, on one hand increasing the attention paid towards vaccination against preventable diseases, but on the other hand leading to several logistical problems connected to vaccine delivery during daily activities. These issues could be associated with finding a suitable site inside a clinical ward that is easy to reach but not in direct contact with clinical units and their patients, or with the difficulty of covering three working shifts.

Moreover, the analysis of reasons for adherence to immunization programs represent a poorly explored element in literature compared with the analysis of refusal motives. Further exploration of this topic might lead to better comprehension of HCWs beliefs and behavior as they pertain to these practices [26,27].

In conclusion, as described in the literature, several factors bear different weight based on the context and times of immunization campaigns. The idea of a unique strategy for every healthcare facility is unrealistic, and so hopefully our positive experience could be a baseline to enforce new ideas and drafts, not only for next year’s campaign but also for other hospitals and healthcare facilities.

## Figures and Tables

**Figure 1 ijerph-18-05874-f001:**
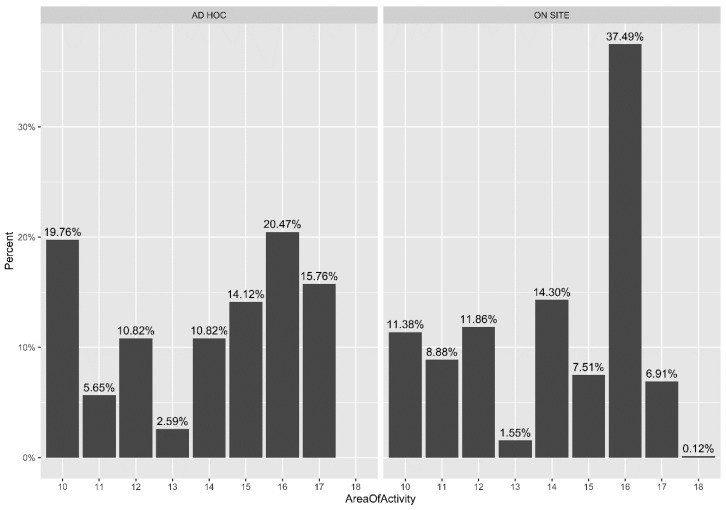
Comparison between area of activity at ad hoc and on-site ambulatory. Area of activity: 10 Administrative; 11 Newborn; 12 Pediatric; 13 General Surgery; 14 Specialist Surgery; 15 General Medicine; 16 Specialist Medicine; 17 Intensive Care Unit; 18 Missing Data.

**Figure 2 ijerph-18-05874-f002:**
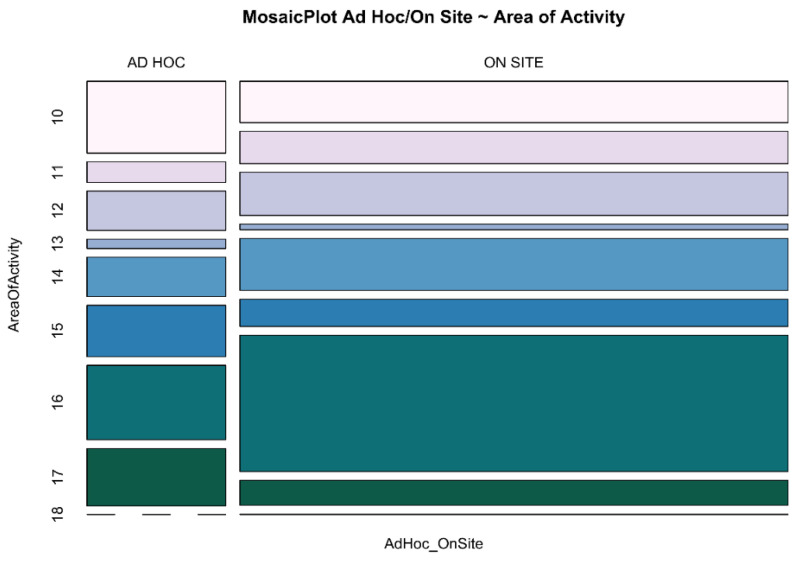
Comparison between area of activity at ad hoc and on-site ambulatory. Area of activity: 10 Administrative; 11 Newborn; 12 Pediatric; 13 General Surgery; 14 Specialist Surgery; 15 General Medicine; 16 Specialist Medicine; 17 Intensive Care Unit; 18 Missing Data.

**Figure 3 ijerph-18-05874-f003:**
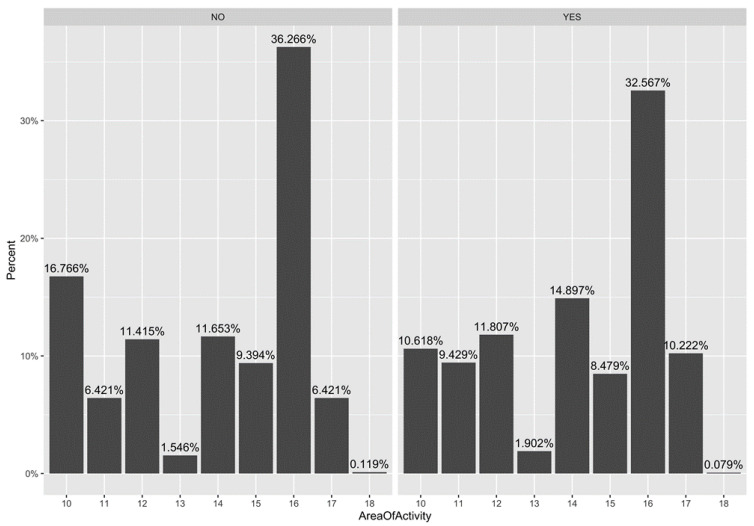
Comparison between area of activity and previous influenza vaccination. Area of activity: 10 Administrative; 11 Newborn; 12 Pediatric; 13 General Surgery; 14 Specialist Surgery; 15 General Medicine; 16 Specialist Medicine; 17 Intensive Care Unit; 18 Missing Data.

**Figure 4 ijerph-18-05874-f004:**
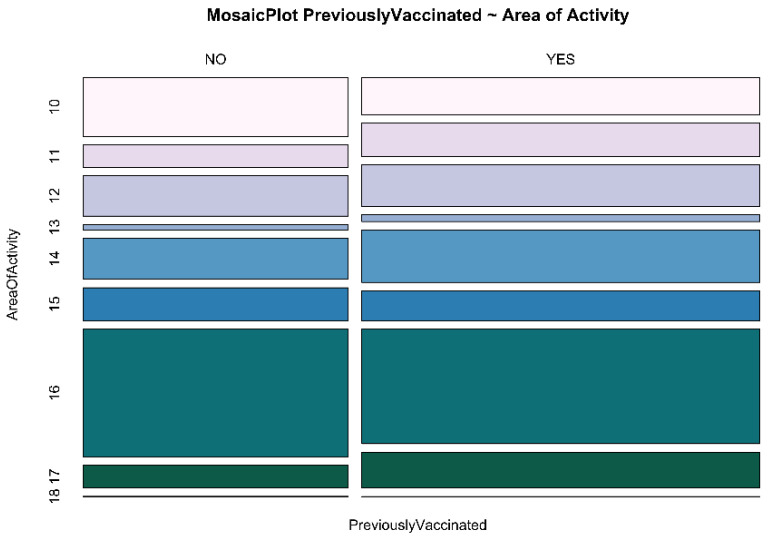
Comparison between area of activity and previous influenza vaccination. Area of activity: 10 Administrative; 11 Newborn; 12 Pediatric; 13 General Surgery; 14 Specialist Surgery; 15 General Medicine; 16 Specialist Medicine; 17 Intensive Care Unit; 18 Missing Data.

**Table 1 ijerph-18-05874-t001:** Comparison between vaccinated populations, 2020–2021 and 2019–2020 seasons.

Features of Vaccinated HCWs	Variation	2020–2021		2019–2020	
Total population vaccinated	+82.4%	2103	(100%)	1153	(100%)
Gender N (%)					
F	+91.6%	1418	(67.5%)	740	(64.2%)
M	+65.9%	685	(32.5%)	413	(35.8%)
Age					
Median, IQR		43.23		36.25	
18–39	+46.1%	938	(44.6%)	642	(55.7%)
40–59	+147.3%	925	(44.0%)	374	(32.4%)
60–80	+75.2%	240	(11.4%)	137	(11.9%)
Occupation					
Physician	+112.0%	600	(28.5%)	283	(24.5%)
Resident	−8.0%	219	(10.4%)	238	(20.6%)
Student	−4.8%	158	(7.5%)	166	(14.4%)
Nurse	+164.3%	452	(21.5%)	171	(14.8%)
Other	+103.0%	203	(9.7%)	100	(8.7%)
Technician	+61.5%	155	(7.4%)	96	(8.3%)
Administrative	+308.3%	196	(9.3%)	48	(4.2%)
Auxiliary staff	+252.9%	120	(5.7%)	34	(2.9%)
Volunteer	/	/	/	16	(1.4%)
NA	/	/	/	1	(0.1%)
Area of Activity					
Administrative	+918.5%	275	(13.1%)	27	(2.3%)
Newborn Area	+78.4%	173	(8.2%)	97	(8.4%)
Pediatric Area	+109.4%	245	(11.7%)	117	(10.1%)
General Surgery	−15.9%	37	(1.8%)	44	(3.8%)
Specs Surgery	+68.2%	286	(13.6%)	170	(14.7%)
General Medicine	−17.7%	186	(8.8%)	226	(19.6%)
Specs Medicine	+91.4%	716	(34.0%)	374	(32.4%)
Intensive Care Unit	+154.2%	183	(8.7%)	72	(6.2%)
Other	/	/	/	15	(1.3%)
NA	−81.8%	2	(0.1%)	11	(1.0%)

**Table 2 ijerph-18-05874-t002:** Prevalence ratio of on-site vs ad hoc ambulatory.

Variable	PR (95% C.I.)	X^2^ Test (Likelihood)
Gender		
Female	Reference	0.76641
Male	0.99 (0.95–1.04)	
Age		
18–39	Reference	0.08831
40–59	1.03 (0.99–1.08)	
60–80	1.07 (1.01–1.15)	
Profession		
Physician	Reference	<0.0001
Resident	1.06 (0.99–1.13)	
Student	0.81 (0.72–0.91)	
Nurse	0.94 (0.88–1.00)	
Other	0.99 (0.66–1.17)	
Technician	0.99 (0.91–1.07)	
Administrative	1.03 (0.96–1.11)	
Auxiliary staff	1.01 (0.92–1.10)	
Area of activity		
Administrative Area	Reference	<0.0001
General Medicine	0.97 (0.85–1.10)	
Newborn Area	1.24 (1.12–1.37)	
Pediatric Area	1.17 (1.06–1.29)	
General Surgery	1.01 (0.81–1.27)	
Specs Surgery	1.21 (1.10–1.33)	
Specs Medicine	1.26 (1.16–1.37)	
Intensive Care Unit	0.91 (1.07–1.81)	
NA	1.44 (1.33–1.56)	

**Table 3 ijerph-18-05874-t003:** Prevalence ratio of never vaccinated before versus vaccinated before 2020.

Variable	PR (95% C.I.)	X^2^ Test (Likelihood)
Gender		
Female	Reference	0.00168
Male	1.12 (1.05–1.21)	
Age		
19–39	Reference	0.20001
40–59	0.96 (0.89–1.04)	
60–80	1.07 (0.96–1.19)	
Profession		
Physician	Reference	<0.000001
Resident	1.12 (1.04–1.20)	
Student	0.59 (0.49–0.70)	
Nurse	0.78 (0.72–0.86)	
Other	0.55 (0.46–0.65)	
Technician	0.59 (0.49–0.71)	
Administrative	0.58 (0.49–0.69)	
Auxiliary staff	0.73 (0.61–0.86)	
Area of activity		
Administrative Area	Reference	<0.000001
General Medicine	1.18 (0.99–1.40)	
Newborn Area	1.41 (1.20–1.65)	
Pediatric Area	1.25 (1.07–1.46)	
General Surgery	1.33 (1.02–1.74)	
Specs Surgery	1.35 (1.16–1.56)	
Specs Medicine	1.18 (1.03–1.35)	
Intensive Care Unit	1.44 (1.24–1.69)	
Other	1.03 (0.25–4.12)	

**Table 4 ijerph-18-05874-t004:** Reasons for vaccination (2020–2021).

Adhesion Data	Total = 1899	
Gender N (%)		
F	1259	(66.3%)
M	601	(31.6%)
NA	39	(2.1%)
Age		
Median	43	±12.55
18–39	784	(41.3%)
40–59	782	(41.2%)
60–80	251	(13.2%)
NA	82	(4.3%)
Occupation		
Physician	524	(27.6%)
Resident	201	(10.6%)
Student	64	(3.4%)
Nurse	351	(18.5%)
Other	338	(17.8%)
Technician	144	(7.6%)
Administrative	179	(9.4%)
Auxiliary staff	78	(4.1%)
Volunteer	/	/
NA	20	(1.0%)
Reasons for vaccination	No. of answers	%
I think vaccination is an effective preventive technique	1437	75.7%
As a HCW I get vaccinated to protect my patients	806	42.4%
As a HCW I am more exposed to influenza	359	18.9%
I work with frail people	295	15.5%
I live with frail people	235	12.4%
I fear influenza complications	157	8.3%
I was convinced by the vaccine presentation campaign	86	4.5%
I fear COVID-19	80	4.2%
Other	58	3.0%
NA	20	1.0%

## Data Availability

Data in Results section are internal data provided by internal service of Department Biomedical Sciences for Health, Postgraduate School in Public Health.

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
