# Peer review of "Influenza Vaccination Campaign during the COVID-19 Pandemic: The Experience of a Research and Teaching Hospital in Milan"

_ijerph, 2021, doi:10.3390/ijerph18115874_

Round 1
Reviewer 1 Report
This is an observational study which was carried out through questionnaires where authors tried to analyse the 2020 influenza vaccination campaign, comparing it with previous one in a research and teaching hospital in Northern Italy. They observed a significant increase of vaccination coverage rate that confirmed the suitability of combined strategy to deliver the flu campaign, representing a first step to reach WHO recommended vaccination rates.
Improvements in terms of language and grammar are necessary.
In the introduction section, a more detailed description, through the relevant references of the importance of the vaccine and the lack of reporting and/or vaccination, is needed, leading to the rationale of the study. Relevant numbers from other countries and nations could be provided, if they exist (there are probably).
Authors should better synchronize their sections, MM and results, that is, what is scheduled in the former, should be analyzed in the latter, so that the reader can more easily follow the flow of the paper.
The discussion section needs reformatting, in terms of a more structured way of reporting. In this context, it should split into more paragraphs. At first, authors should report on the main results of their study, then compare them their results with similar studies and analyzing all their findings, then reporting on the limitations of their study.
The reference list should be enriched.
Author Response
Dear Reviewer,
I would like to thank you for your comments and suggestions which have shown us through a deep review process that changed the article. I uploaded the new version with adjustments marked through track changes. Furthermore, I’ll try to explain the changes linked to your suggestions point-by-point.
This is an observational study which was carried out through questionnaires where authors tried to analyse the 2020 influenza vaccination campaign, comparing it with previous one in a research and teaching hospital in Northern Italy. They observed a significant increase of vaccination coverage rate that confirmed the suitability of combined strategy to deliver the flu campaign, representing a first step to reach WHO recommended vaccination rates.
Improvements in terms of language and grammar are necessary.
During the review process we tried to improve language, according to your suggestion.
In the introduction section, a more detailed description, through the relevant references of the importance of the vaccine and the lack of reporting and/or vaccination, is needed, leading to the rationale of the study. Relevant numbers from other countries and nations could be provided, if they exist (there are probably).
We have reported vaccination coverage rate of several European countries; we have choose not to report numbers from world-wide countries to avoid bias elements linked to cultural change or mandatory vaccination policies. We also introduce a brief comment on economic implications of influenza vaccination and its refusal.
Authors should better synchronize their sections, MM and results, that is, what is scheduled in the former, should be analyzed in the latter, so that the reader can more easily follow the flow of the paper.
MM section was reformatted and enriched in order to reach a better comprehension and a simpler analysis. Results section was reorganized better to match with MM section and to simplify the reader follow the flow of the paper.
The discussion section needs reformatting, in terms of a more structured way of reporting. In this context, it should split into more paragraphs. At first, authors should report on the main results of their study, then compare them their results with similar studies and analyzing all their findings, then reporting on the limitations of their study.
We have reformatted the discussion section, according to your suggestion; moreover, we have described the several limitations of the study.
The reference list should be enriched.
We have enriched the reference list increasing of 22% the number of references.
Reviewer 2 Report
Helpful, if limited study of influenza vaccination among healthcare workers. Contains significant interesting information with respect to increase of adherence to vaccination among HCW and associated personnel. May be published after minor editorial revision with respect to English (e.g., 'physicians were represented to a higher level', line 143; 'this percentage being', line 225; 'explained by', line 226; and a few others), please make sure not to switch from present tense to past tense when describing results. Also the manufacturer of Flucelvax Tetra® (line 98) should be indicated.
Author Response
Dear Reviewer,
I would like to thank you for your comments and suggestions which have shown us through a deep review process that changed the article. I uploaded the new version with adjustments marked through track changes. Furthermore, I’ll try to explain the changes linked to your suggestions point-by-point.
Helpful, if limited study of influenza vaccination among healthcare workers. Contains significant interesting information with respect to increase of adherence to vaccination among HCW and associated personnel.
We thank you very much for your comments, specially for the significant interesting information.
May be published after minor editorial revision with respect to English (e.g., 'physicians were represented to a higher level', line 143; 'this percentage being', line 225; 'explained by', line 226; and a few others), please make sure not to switch from present tense to past tense when describing results.
During the review process we tried to improve language, according to your suggestion.
Also the manufacturer of Flucelvax Tetra® (line 98) should be indicated.
We have already change according to your suggestion.
Reviewer 3 Report
The article presents a very brief introduction.
The methodology should be better structured and better explained.
The results are little explored and at the same time with too many tables and figures that could be condensed. With the data presented, the discussion could be richer in addition to not presenting any limitations of the study.
Author Response
Dear Reviewer,
I would like to thank you for your comments and suggestions which have shown us through a deep review process that changed the article. I uploaded the new version with adjustments marked through track changes. Furthermore, I’ll try to explain the changes linked to your suggestions point-by-point.
The article presents a very brief introduction.
The introduction has been extended reporting a description of vaccine coverage rates in several European countries, in order to show similarity and difference between different populations. Moreover, we have introduce a brief comment on economic implications of influenza vaccination and its refusal.
The methodology should be better structured and better explained.
We have restructured and enriched the methodology by adding a more detailed description of ad ho and on site vaccination ambulatory.
The results are little explored and at the same time with too many tables and figures that could be condensed.
We have explained the reasons for analyzing features referring to some specific elements, such as area of activity, instead an analyzing all elements. We also have enriched the results with a more detailed description of vaccine adherence questionnaire.
With the data presented, the discussion could be richer in addition to not presenting any limitations of the study.
We have introduced the several limitation of the study and also we have reviewed the discussion to make easier for reader the comprehension of paper’s flow.
Reviewer 4 Report
The work is very well described, and this type of vaccination campaign should be done in all hospitals.
I have just a few observations and suggestions:
Why was only the previous season 2019-2020 taken into consideration?
Line 152: I would insert "season 2020-2021" and in the legend instead of the abbreviation H and O, I suggest writing ad hoc and on site of figure 1.
Line 158: same observation as line 152.
Line 171: Does previous vaccination match "yes" and first vaccination match "no" in Figures 3 and 4?
In figures 3 and 4 change SI to YES.
Author Response
Dear Reviewer,
I would like to thank you for your comments and suggestions which have shown us through a deep review process that changed the article. I uploaded the new version with adjustments marked through track changes. Furthermore, I’ll try to explain the changes linked to your suggestions point-by-point.
The work is very well described, and this type of vaccination campaign should be done in all hospitals.
We thank you very much for your awesome praise, specially for your proposal of implementation of our campaign strategies in all hospital.
I have just a few observations and suggestions:
Why was only the previous season 2019-2020 taken into consideration?
We have taken into consideration this single season because it was the first campaign in which we have implemented the ad hoc vs on site vaccination strategy. We wanted to analyze primarily the implication of the new strategies introduce for the first time, the advertising campaign and the departments challenge, so in order to avoid bias connected to a different vaccination supply we performed comparison only with 2019-2020 vaccination campaign.
Line 152: I would insert "season 2020-2021" and in the legend instead of the abbreviation H and O, I suggest writing ad hoc and on site of figure 1.
We have already change according to your suggestion.
Line 158: same observation as line 152.
We have already change according to your suggestion.
Line 171: Does previous vaccination match "yes" and first vaccination match "no" in Figures 3 and 4?
Yes, previous vaccination match “yes”, never vaccination before or first vaccination during 2020-2021 campaign match “no”.
In figures 3 and 4 change SI to YES.
We have already change according to your suggestion.
Round 2
Reviewer 3 Report
Resultados da tradução
The authors followed the recommendations given.